# The State of the Dopaminergic and Glutamatergic Systems in the Valproic Acid Mouse Model of Autism Spectrum Disorder

**DOI:** 10.3390/biom12111691

**Published:** 2022-11-15

**Authors:** Alexandre Maisterrena, Emmanuel Matas, Helene Mirfendereski, Anais Balbous, Sandrine Marchand, Mohamed Jaber

**Affiliations:** 1Laboratoire de Neurosciences Expérimentales et Cliniques, Inserm, Université de Poitiers, 86000 Poitiers, France; 2Pharmacologie des Agents Anti-Infectieux et Antibiorésistance, Inserm, Université de Poitiers, 86000 Poitiers, France; 3CHU de Poitiers, 86000 Poitiers, France

**Keywords:** dopamine, glutamate, autism spectrum disorder, valproic acid, mouse models, striatum, cerebellum, accumbens

## Abstract

Autism Spectrum Disorder (ASD) is a progressive neurodevelopmental disorder mainly characterized by deficits in social communication and stereotyped behaviors and interests. Here, we aimed to investigate the state of several key players in the dopamine and glutamate neurotransmission systems in the valproic acid (VPA) animal model that was administered to E12.5 pregnant females as a single dose (450 mg/kg). We report no alterations in the number of mesencephalic dopamine neurons or in protein levels of tyrosine hydroxylase in either the striatum or the nucleus accumbens. In females prenatally exposed to VPA, levels of dopamine were slightly decreased while the ratio of DOPAC/dopamine was increased in the dorsal striatum, suggesting increased turn-over of dopamine tone. In turn, levels of D1 and D2 dopamine receptor mRNAs were increased in the nucleus accumbens of VPA mice suggesting upregulation of the corresponding receptors. We also report decreased protein levels of striatal parvalbumin and increased levels of p-mTOR in the cerebellum and the motor cortex of VPA mice. mRNA levels of mGluR1, mGluR4, and mGluR5 and the glutamate receptor subunits NR1, NR2A, and NR2B were not altered by VPA, nor were protein levels of NR1, NR2A, and NR2B and those of BDNF and TrkB. These findings are of interest as clinical trials aiming at the dopamine and glutamate systems are being considered.

## 1. Introduction

Autism spectrum disorder (ASD) is a neurodevelopmental disorder affecting an increasing proportion of the population and that is mainly characterized by the presence of social deficits and restricted and repetitive behavior and interests [1]. Management of this disorder mainly targets associated comorbidities while effective pharmacological therapies for the core symptoms remain unavailable [2,3]. This is in great part due to ASD’s etiology and physiopathology that are still at an early stage of scientific discovery. Over a thousand ASD-related genes have been identified and they mainly target neurotransmission systems at the levels of synapses [4,5]. Exposure of pregnant mothers to several environmental factors has also been described as increasing the risk of developing ASD in the offspring [6,7,8,9]. The environmental factors are numerous, among which the most documented are pharmacological (Depakine, through its active ingredient, valproic acid, VPA) [10] or infectious (inducing maternal immune activation, MIA) [11].

Despite the high heterogeneity in ASD symptoms and etiology, the occurrence of shared symptoms suggests common deficits in common neurodevelopmental pathways [12,13,14,15]. Given that neurotransmitter systems play a major role in shaping the nervous system during development, a dysfunction at this level may contribute to the pathophysiology of ASD. Indeed, normal brain development and functioning depends in large part on the physiological equilibrium of neurotransmission systems. An imbalance in the glutamate and dopamine neurotransmission systems is thought to be a main cellular and molecular component of ASD pathophysiology [14,15]. Disturbances in these neurotransmission systems during development can originate from a myriad of genetic and/or environmental factors and may converge to the common behavioral symptoms that are the hallmarks of ASD. Determining the biological basis of ASD is of relevance as it may lead to better diagnosis and potential pharmacological therapies for a psychiatric condition that is in dire need.

Many genes within the glutamatergic system have been associated with ASD. These include, but are not limited to, *NRX1-3* (neurexin 1-3), *NLGN1, 3, 4* (neuroligin1, 3, 4), *CNTNAP2* (contactin-associated protein 2), and *SHANK 1-3* (SH3 and multiple ankyrin repeat domains proteins) [14,16,17]. However, whether ASD patients and animal models present a hyper or a hypoglutamatergic state is still debatable as two opposed theories have been proposed by Carlsson et al. (1998) [18] and Fatemi et al. (2008) [19], respectively. On the one hand, high levels of glutamate were reported in ASD patients’ blood [20] and were linked to ASD severity [21,22]. In the VPA animal model, the NMDA-R (N-methyl-D-aspartate receptor) subunits GluN2A and GluN2B levels were upregulated, as were AMPA-R (a-amino-3-hydroxyl-5-methyl-4-isoxazole-propionate receptor) subunits GluA1 and GluA2 levels, while AMPA-R antagonists ameliorated social deficits [23]. On the other hand, a hypoglutamatergic state in ASD is suggested in clinical findings where a positive AMPA modulator was shown to reduce some of the ASD symptoms [24] and also in ASD animal models where restoring the NMDA function was shown to improve social parameters [25] (reviewed in [26]). These discrepancies may stem from the wide spectrum of the disorder in terms of not only its environmental and genetic etiology, but also its phenotypic expression. The main hypothesis on the pathophysiology of ASD stipulates that there is an excitation/inhibition (E/I) imbalance underlying the disorder and implicating the glutamatergic and GABAergic systems [27]. This, along with the fact that several genes implicated in ASD are major actors in glutamatergic transmission [14], has spurred a number of clinical trials and experimental assays aimed at targeting the glutamatergic neuronal transmission.

Dopamine plays a critical role in motivated behavior, action selection, and reward processing [28]. In addition, dopamine is also a major actor in social function as it is released in response to social cues during social seeking behavior [29]. Positron emission imaging studies in ASD patients have shown a significant reduction in prefrontal cortical levels of dopamine [30] and more recent MRI studies have shown several alterations in dopaminergic connectivity within the caudate nucleus in relation to ASD [31,32,33]. Interestingly, the extent of these alterations has been correlated with the severity of ASD symptoms [34] (also reviewed in [35]).

Beyond imaging studies, post-mortem studies are hindered by the limited availability of brain samples and tissues from ASD patients. Conveniently, ASD animal models, whether genetic or environmental, have regularly been proven to present a strong construct and face validity, with some showing promise for their predictive validity [36,37,38,39]. We have recently reported findings along this line, with genetic (Shank3) and environmental (VPA and MIA) ASD models [10,11,40]. To various degrees, these models reproduced the spectrum of the disease as they all showed major behavioral alterations that were reminiscent of ASD core symptoms, with clear-cut social deficits and stereotyped behaviors. These alterations were also accompanied by gait and motor deficits and were correlated with loss of Purkinje cells (PC) within restricted cerebellar subregions. Additionally, we reported alterations in the synaptic structures of striatal medium spiny neurons as well as in their excitability [41].

Here, we have focused on determining the state of several key players of the dopamine system within the ventral mesencephalon, the striatum, the nucleus accumbens, and the frontal and motor cortex. We have also examined the expression of several glutamate receptors in these brain regions as well as of Brain-Derived Neurotrophic Factor (BDNF) and neurotrophic tyrosine receptor kinase beta (TrkB). This was performed by stereological counting of dopamine cells, analysis of dopamine levels with mass spectrometry, quantitative polymerase chain reaction (qPCR) and membrane blotting. Studies were performed in the VPA animal model of ASD as previously described [10] and in which VPA was administered as a single dose (i.p., 450 mg/kg) to pregnant females at E12.5 when the neuronal tube is closing in rodents, followed by neurogenesis and neuronal migration [42]. In this model, we previously reported that social behavior was dramatically impaired in male, but not female, VPA pre-exposed mice [10]. Additionally, we found that both males and females have significant impaired gait and motor coordination. At the cellular level, we found a specific and restricted decrease in the number of cerebellar PC and that was in the Crus I subregion in males and in Crus II in females. We performed correlation analyses and reported that motor impairments strongly correlate to both social deficits and to the number of cerebellar PC.

## 2. Materials and Methods

### 2.1. Experimental Design

C57BL/6J Mice (Charles Rivers, Lyon, France) were housed at the local Prebios animal facility, in ventilated cages with ad libitum access to food and water. Room temperature was maintained at 23 °C on a 12 h light/dark cycle.

All animals used here had previously undergone full-scale behavioral analysis before being sacrificed. For this, 37 females and 23 males were used for mating. Three females were placed with a single male and left for 3 days. Males were retrieved from the cage following mating as evidenced by the presence of a vaginal plug. Pregnant mice received a single i.p. injection of either VPA (450 mg/kg) or NaCl 0.9% at gestational day 12.5 (E12.5) when the neuronal tube is closing in rodents, followed by neurogenesis and neuronal migration [10,37]. Following mating, pregnant mice were left undisturbed until they gave birth. At weaning on postnatal day 21 (P21), sex and age-matched pups were separated and raised by groups of 4 in a randomized fashion to avoid littermate effects. All mice were identified by ear tags. The experiment timeline was provided in our initial paper [10]. Following behavioral analysis, animals were assigned to different experimental groups and were sacrificed. Brains were harvested for further analysis as described below. The experimenter was blind to the treatment until all experiments and analyses were completed.

### 2.2. Tyrosine Hydroxylase (TH) Immunohistochemistry at the Ventral Mesencephalon Level

Male mice (saline (*n* = 11) and VPA (*n* = 7)) were deeply anesthetized with ketamine-xylazine (120–20 mg/kg) and transcardially perfused with 0.9% saline at 37 °C followed by 4% paraformaldehyde (PFA) at 4 °C. Brains were post-fixed in 4% PFA at 4 °C for 24 h before cryoprotection in 30% sucrose for 48 h. Serial 50 µm free-floating sections were collected and stored in an anti-freeze solution at −20 °C until use. Every fourth section was mounted on gelatin-coated slides for quantification of TH-positive neurons that were further identified based on their morphology and localization. Stereological estimates were performed using the optical fractionator method and systematic random sampling to obtain the total number of TH-positive neurons. Neurons were counted at 40 × objective using the Mercator image analysis system (Explora Nova, La Rochelle, France). Upper and lower guard zones of 1 µm were set at the top and bottom of the section to exclude lost profiles and each neuron or visible nucleus was counted as previously described [11].

### 2.3. Chromatography and Spectrometry Analysis

Mice were deeply anesthetized with ketamine–xylazine (120–20 mg/kg) and transcardially perfused with 1X PBS at 37 °C. Brains were retrieved, regions of interest were either dissected out (cerebellum) from the frozen brain or through punches (striatum, nucleus accumbens, mesencephalon) and were frozen in −80 °C until use.

High-performance liquid chromatography (HPLC) (5200a Coulochem III, ESA, Chelms-ford, MA, USA) was used to quantify dopamine levels within the striatum and accumbens of mice (saline (*n =* 12) and VPA (*n =* 10)). Punches were sonicated in 100 uL distilled water 30 s and centrifugated 10 min at 10,000× *g.* Supernatant was collected and used in the HPLC device as previously described [43]. The mobile phase contained 100 mM NaH_2_PO_4_, 0.1 mM Na_2_EDTA, 0.5 mM n-octyl sulfate, and 18% (*v/v*) methanol (pH adjusted to 5.5 with Na_2_HPO_4_). Assay sensitivity for dopamine was 2 fmoles per sample. Data are expressed as pg of DA/ug of proteins.

Ultra-performance liquid chromatography–tandem mass spectrometry detection was used to quantify dopamine and 3 of its metabolites, dihydroxyphenylacetic acid (DOPAC), homovanillic acid (HVA) and 3-methoxytyramine (3-MT) in striatal regions and nucleus accumbens following protein precipitation by acetonitrile (male saline (*n =* 18), male VPA (*n =* 20), female saline (*n =* 18), female VPA (*n =* 18)). The equipment was composed of a quadrupole tandem mass spectrometer (QT3500, Sciex, Framingham, MA, USA) and HPLC (Nexerra XR-LC system, Schimadzu, Kyoto, Japan) with a data acquisition station (Analyste version 1.6.3). The analytical column was a Luna^®^ omega Polar C18 column (100Å, 5 µm, 2.1 × 150 mm) (Phenomenex, Torrance, CA, USA). The mobile phase consisted of a mixture of water and acetonitrile (ACN) using a gradient mode at 30 °C with a flow of 0.2 mL/min. For MS/MS detection, electrospray ionization (ESI) was used in positive mode for dopamine and 3 MT, and in negative mode for DOPAC and HVA. Mass spectra were acquired by multiple reaction monitoring. The specific transition used for quantification was 153.98 to 91.1 (Dopamine), 166.97 to 123.0 (DOPAC), 180.99 to 121.9 (HVA), 168.0 to 119.0 (3-MT). For the internal standards (IS), the transition was 153.1 to 95.0 (Dopamine IS), 171.0 to 128 (DOPAC IS), 187.02 to 142.7 (HVA IS) and 172.1 to 94.9 (3-MT IS). For all compounds, range concentration was between 10 to 5000 ng/mL. IS of dopamine, DOPAC, HVA and 3-MT were [2H4]-Dopamine, [2H5]-dihydroxyphenyl acetic acid, [13C6]-homovanillic acid, [2H4]-3-methoxytyramine, respectively. Within run accuracy and precision were evaluated by assaying the quality control (QC) samples with three concentration levels (low, medium, and high) three times in a single analytical run while the between run accuracy and precision were performed by assaying the QC samples with tree concentration levels four times in at least four separate days. Accuracy error was expressed as a bias and the precision was expressed as the coefficient of variation (CV). The acceptance criteria for QCs were met for all compounds during the analytical analysis (Bias = ± 15% and CV ≤ 15%). Experiments were performed under ISO 9001 procedures. Data are expressed as pg of DA/ug of proteins.

### 2.4. Quantitative Immunoblot Analysis

Male mice (saline (*n =* 14) and VPA (*n =* 14) in total) were sacrificed, brains were retrieved, regions of interest were either dissected out from the frozen brain (cerebellum: saline 1 to 5 (*n =* 5) and VPA 1 to 5 (*n =* 5)) or through punches (striatum: saline 1 to 9 (*n =* 9) and VPA 1 to 9 (*n =* 9), nucleus accumbens: saline 10 to 14 (*n =* 5) and VPA 10 to 14 (*n =* 5), motor cortex: saline 1 to 9 (*n =* 9) and VPA 1 to 9 (*n =* 9)). Proteins were extracted using 1% sodium dodecyl sulfate (SDS) solution in Tris HCl 0.1M with ethylenediaminetetraacetyl (EDTA) 0.01M and phenylmethylsulfonyl fluoride (PMSF), protease inhibitor, and phosphatase inhibitor at 1%. Equal amounts of proteins from lysates were separated by SDS-PAGE and migrated proteins were transferred to nitrocellulose membranes (Bio-Rad, Marnes-la-Coquette, France). After blocking the membranes at room temperature for 1 h in Tris Buffer Solution with Tween-20 0.1 M (TBST) at pH 7.4 and 5% non-fat milk, blots were incubated with corresponding primary antibodies at room temperature for 3 h. The blots were then washed three times in TBST and incubated with HRP-conjugated secondary antibodies overnight at 4 °C. Following 3 washes in TBST, the blots were incubated with ECL reagent (GE Healthcare Life Sciences, Chicago, IL, USA). For quantification, the films were scanned by a PXi image system and gray signals were analyzed by GeneTools software (Syngene, Cambridge, UK), and normalized to that of corresponding internal controls (actin or α-tubulin). The following primary antibodies were used at the dilution of 1/500 to 1/1500: TH, BDNF, TrkB/TrkB-T1, LC3-I/II, DARPP32, p-DARPP-32, mTOR 7C10, P-mTOR ser 2448, ERK1/2, and P-ERK1/2 (Cell Signaling Technology, Leiden, The Netherlands), parvalbumin, NR2A, NR2B (Millipore, Molsheim, France), and NR1 (Antibodies Incorporated, Davis, CA, USA). A-tubulin (1/10,000) and β-actin were used at a (1/10,000) dilution (Sigma-Aldrich, Saint Quentin Fallavier, France).

### 2.5. QPCR Analysis

Male mice (saline (*n =* 9) and VPA (*n =* 8)) were sacrificed, brains were collected, and striatal, cerebellar, and cortical regions were dissected out and frozen in −80 °C until use. Total RNA was isolated using TRIzol Reagent/chloroform, then purified using a NucleoSpin RNA kit (Macherey-Nagel, Hœrdt, France) and quantified using NanoDrop ND-1000 spectrophotometer (Thermo Fisher Scientific, Waltham, MA, USA). RNA integrity was evaluated by an Agilent 2100 Bioanalyzer System. All RNAs had an RNA integrity number above 7. Reverse transcription was performed on 1μg of total RNA for each sample using Verso cDNA Synthesis kit (Thermo Fischer Scientific, Waltham, MA, USA). qPCR was performed on LightCycler 480 system (Roche Diagnostics, Meylan, France). Ct values were averaged from triplicates. Results were subtracted to the mean Ct of the housekeeping gene *Gapdh* (ΔCt) and log transformed (2^−ΔCt^). Primer sequences are provided in Appendix A.

### 2.6. Data Analysis

Data are expressed as mean ± Standard Error of the Mean (SEM) and analyzed using GraphPad Prism-7 software (La Jolla, San Diego, CA, USA) unless otherwise indicated in the figure legends. Immunoblots, qPCR and histological measures that followed a normal distribution were analyzed using Student’s *t*-test. Spectrophotometry measures were analyzed using Mann–Whitney non-parametric tests. For all analyses, a *p* value of < 0.05 was considered significant.

## 3. Results

### 3.1. Dopamine Neurons and Tyrosine Hydroxylase Levels Are Not Altered by VPA Treatment

We have previously shown that mice pre-exposed to VPA (450 mg/kg) at E 12.5 showed major social deficits [10], in line with clinical observations [44]. Given the described role of accumbal dopamine in modulating social interactions, we aimed here at investigating the number of dopamine cells within the ventral mesencephalon and the corresponding dopamine levels within target areas. We found that the number of dopamine neurons, as evidenced by TH+ labelling, was not altered by VPA (Figure 1A). Accordingly, TH protein levels were not altered in either the striatum or the nucleus accumbens (Figure 1B). These results suggest that dopamine transmission might not be altered within the mesolimbic and the nigrostriatal pathway.

### 3.2. Dopamine and Metabolites Levels Are Altered by VPA Exposure

Given the stable number of dopamine neurons and TH protein levels within the mesolimbic and nigrostriatal pathways, we aimed to measure dopamine and metabolite levels within these pathways. We first measured dopamine levels by HPLC in saline and VPA male mice and found no alteration within the striatum or the nucleus accumbens (Figure 2A). We next explored levels of dopamine and metabolites within the ventral mesencephalon (Mes), the nucleus accumbens (Nac), the whole striatum (CPU), and the dorsal striatum (CPU dorsal) using the more sensitive mass spectrometry procedure in separate groups of male and female mice (Figure 2B,C). We found that female VPA mice showed a slight decrease in the levels of dopamine within the dorsal CPU (*p* < 0.05) associated with an increase in the DOPAC/DA ratio (*p* < 0.05) that generally indicates a higher turnover rate of dopamine metabolism. No other alteration in the levels of dopamine, DOPAC, HVA, and 3-MT in any of the brain regions explored was found.

These results indicate slight but significant decreases in dopamine levels suggesting a potential role of dopamine in social deficits and motor disorders reported in VPA mice [10].

### 3.3. Dopamine and Glutamate Receptors’ Levels in Response to VPA Exposure

We measured mRNA levels of major dopamine and glutamate receptors within target brain areas that are suspected to be implicated in major ASD social and motor symptomatology: the nucleus accumbens (Figure 3A), the striatum (Figure 3B), the motor cortex (Figure 3C), and the cerebellum (Figure 3D). We found a two-fold increase in mRNA levels of D1 and D2 dopamine receptors (DRD1 and DRD2 respectively) within the nucleus accumbens (Figure 3A). This could indicate an upregulation of the corresponding receptors as a compensation for the observed decrease in dopamine levels.

Glutamate is a major excitatory neurotransmitter that regulates cognitive functions such as attention, learning, and memory, which are affected in ASD. Indeed, many of the genes implicated in ASD are related to glutamatergic transmission and synapses. We thus investigated levels of key glutamatergic receptors and receptor subunits in response to VPA prenatal exposure. We found no alteration in the mRNA levels of *NR1, NR2A, NR2B, mGluR1, mGluR4,* and *mGluR5* in any of the brain regions examined (nucleus accumbens, striatum, motor cortex, and cerebellum) (Figure 3). Additionally, there were no alterations in the protein levels of NR1, NR2A, and NR2B in these brain regions (Figure 4), as previously reported [40].

### 3.4. Signal Transduction Pathways within the Dopaminergic and Glutamatergic Systems

We undertook further analysis of major molecules implicated in the dopamine signaling in main dopaminergic brain regions by immunoblotting (Figure 5). We analyzed levels of dopamine- and cAMP-regulated phosphoprotein 32kDa (DARPP-32), considered as a major integrator of dopamine transmission [45]. We report here no alterations in DARPP 32 and p-DARPP 32 in either the nucleus accumbens, striatum, or motor cortex (Figure 5).

We also analyzed levels of TrkB and its truncated isoform TrkB-T1 (Figure 4) and of phosphorylated and non-phosphorylated Erk1 and 2, mTOR and Akt as these receptors and signal transduction effectors are highly regulated in relation with neuronal plasticity and thus are recognized indicators of alterations in neuronal signaling [46]. We found increased levels of p-mTOR in the motor cortex (Figure 5C) and the cerebellum (Figure 5D) but no further alteration in the levels of other proteins investigated whether in their phosphorylated or non-phosphorylated forms within the striatum, the nucleus accumbens, the motor cortex, or the cerebellum (Figure 5). We also investigated levels of striatal parvalbumin (PV) as PV interneurons provide the strongest inhibitory inputs to striatal medium spiny neurons (MSNs) and are in charge of the feed forward inhibition. In addition, PV levels were decreased in ASD patients [47] and in genetic ASD animal models [48]. We confirm these findings and report a slight decrease in PV striatal protein levels in the VPA ASD mouse model (Figure 5B).

## 4. Discussion

Given the major implications of the dopaminergic and glutamatergic neurotransmission systems in the pathophysiology of ASD, we aimed to determine the state of various players in these systems in the VPA animal model. As a prerequisite, we have previously characterized this model at the behavioral and cellular levels and reported major deficits in social activities, gait, and motor coordination. These deficits were directly correlated to the severity of PC loss within the cerebellum [10].

Among the alterations that we found here following prenatal exposure to VPA are (i) slight decreases in the levels of dopamine within the dorsal striatum in females accompanied by (ii) increases in mRNAs coding for the D1 and D2 dopamine receptors within the nucleus accumbens. This suggests that dopamine transmission may be downregulated following VPA prenatal exposure with corresponding upregulation of the two major dopamine receptors. (iii) No alteration was detected in the mRNA and protein levels of several glutamate receptors and subunits. (iv) At the signal transduction level, we report increases in p-mTOR levels in the motor cortex and the cerebellum. (v) We found decreased striatal PV levels in this VPA animal model as also reported in genetic ASD models [48].

Sizable evidence points to a possible dysfunction within the dopamine system that may underlie ASD symptomatology [15,35]. Dopamine is the major catecholamine of the central nervous system and is involved in the regulation of a variety of functions, including locomotor activity and emotion, and affect motivated behavior and reward processing, all of which are affected in ASD. Indeed, alterations in the dopamine system are directly linked to neurological and psychiatric disorders such as Parkinson’s disease and schizophrenia [49,50]. The link between the dopamine system and ASD stems from several observations and can be traced back to the seminal work of Antonino Damasio’s team in the late 1970s describing a neurological model of childhood autism [51]. Based on analogies and signs in adult neurology, the authors proposed that ASD would result from dysfunction in several brain areas including the striatum, raising the possibility that dysfunction of the dopamine system is related to ASD core symptoms. Later on, imaging studies provided consistent findings indicating alterations in the dopamine system in ASD patients. These include the reduction of fluorodopa accumulation [30], abnormalities in dopaminergic structures and connectivity as revealed by MRI [31,32,52], enlargement of caudate volume [53,54], and reduction in grey matter within the fronto–striatal network [55]. These alterations have at times been correlated with the severity of ASD symptoms [35]. The reported clinical evidence was also mirrored in various ASD animal models. For instance, several genetic mutations related to ASD were performed in the Drosophila such as the FMR1 mutation mimicking the Fragile X Syndrome, which is the most frequent monogenic mutation associated to ASD [56]. Authors have reported decreased synthesis of brain dopamine and elevated levels of vesicular monoamine transporter mRNA, which is implicated in the vesicular release of dopamine. The dopamine transporter (DAT) is a major element regulating dopamine transmission [50]. Inactivation of DAT in the Drosophila induces behavioral abnormalities related to ASD [57]. In the Zebrafish, social interaction was shown to be directly related to maturation of the dopamine system [58] while D1 antagonists reduced social behavior [59]. In mice, several reports have also linked dopamine to ASD. For instance, Mecp2-null mice, a mouse model of Rett syndrome with several ASD-related behaviors, show abnormalities in dopamine synapses and a decreased number of TH+ neurons within the mesencephalon [60]. Shank 3b-null mice have an enlargement of the striatum and morphological alterations of the striatal medium spiny neurons [61].

Although consistent, these sets of findings do not allow for determining whether the alterations reported in the dopamine system constitute the molecular and cellular basis of ASD or whether they are merely a consequence of alterations in other systems, which may or may not be related to neuronal transmission. In our hands, and in the VPA mouse model where a single dose of 450 mg/kg VPA was injected at E12.5, we found slight but significant alterations within the dopamine system, indicating a down regulation of dopamine transmission. In addition, we report increased levels of striatal p-mTOR, an integrator of molecular pathways that has been associated with neurodevelopmental, psychiatric, and neurodegenerative disorders including ASD. m-TOR is thought to be an important regulator of striatal functions through an intricate mechanism involving RhoA and as such is considered as a potential therapeutic target [62]. Furthermore, we confirm decreased levels of striatal PV that was previously reported in PV and Shank ASD mouse models [48]. This of interest as a PV hypothesis of ASD was recently formulated based on the facts that (i) PV function is often altered in psychiatric disorders including ASD, (ii) PV mRNA levels are the most down regulated transcripts in human ASD samples, and (iii) the number of PV+ neurons are decreased in ASD brains in patients and animal models (reviewed in [63]).

As with dopamine, early studies of ASD patients indicated alterations in the glutamatergic system [18,19]. Blood levels of glutamate in ASD patients were found to be increased while glutamine levels were decreased [20,22] and these alterations were correlated with the severity of ASD symptoms [21]. Glutamate/glutamine ratio was also found to be increased in several brain regions of ASD patients as measured by magnetic resonance spectroscopy [64,65] although other studies, contradictorily, have reported a decrease in these levels [66]. However, perhaps the most compelling evidence of implication of glutamate in ASD arises from genetic studies. Indeed, amongst the hundreds of candidate genes related to ASD, several seem to be directly associated with glutamate transmission [17]. These include genes encoding for glutamate receptor subunits, receptor regulatory subunits, glial glutamate transporters, and several signal transduction and scaffolding proteins including members of the shank family coding for proteins associated with glutamate receptors [67]. In addition, several studies on ASD animal models have reported an E/I imbalance that may reflect changes in the GABA and/or glutamate transmission [41]. This E/I imbalance may be due to alterations in excitatory or inhibitory currents and is thought to be directly related to ASD-related behavior. In our hands, and in the VPA ASD animal model, we did not detect any alteration in mRNA or protein levels of several metabotropic or ionotropic glutamate receptors or receptor subunits. This raises the possibility that the reported E/I imbalance may not be directly due to levels of glutamate receptors but would potentially result from alterations in GABA receptors expression. The function of associated inotropic receptors regulating membrane excitability and currents could also be altered and deserves further investigations

## 5. Conclusions

The quest to identify molecular targets to ASD is of major importance as no pharmacological management of this neurodevelopmental disorder other than symptomatic is available yet. In accordance with this, our set of results seem to indicate that dopamine and the glutamatergic systems may be only indirectly implicated in the pathophysiology and the subsequent symptomatology of ASD. Given the environmental and genetic heterogeneity of ASD etiologies, a myriad of systems could be potentially affected. These systems may converge towards a cluster of players implicating dopamine and glutamate systems leading to the spectrum of the disease. As such, this does not necessarily translate into alterations of major components of these neurotransmission systems but would rather result in a global disequilibrium away from the physiological homeostasis. If true, the quest for a single regional, cellular, or molecular target to manage all ASD symptoms in all patients might not be fruitful without prior clear identification of the nature of alterations that might be different from one patient to another, in relation with ASD etiology.

## Figures and Tables

**Figure 1 biomolecules-12-01691-f001:**
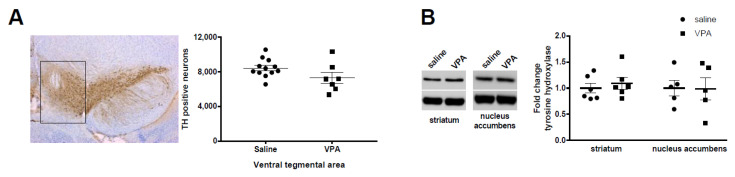
Dopamine neuron numbers and tyrosine hydroxylase levels are not altered by VPA prenatal exposure. (**A**) No significant difference is observed in TH+ cell number within the ventral mesencephalon in response to VPA prenatal exposure. Saline males (*n =* 12); VPA males (*n =* 7). On the left of panel A is a representative picture showing the area (black rectangle) where TH positive neurons were quantified. (**B**) No alterations were found in striatal and accumbal TH levels. Striatum: saline males (*n =* 6), VPA males (*n =* 6) and nucleus accumbens: saline males (*n =* 5), VPA males (*n =* 5). On the left of panel B is a representative immunoblot showing TH (top) and tubulin (bottom) bands. All data are expressed as means ± SEM; Student’s *t*-tests were performed.

**Figure 2 biomolecules-12-01691-f002:**
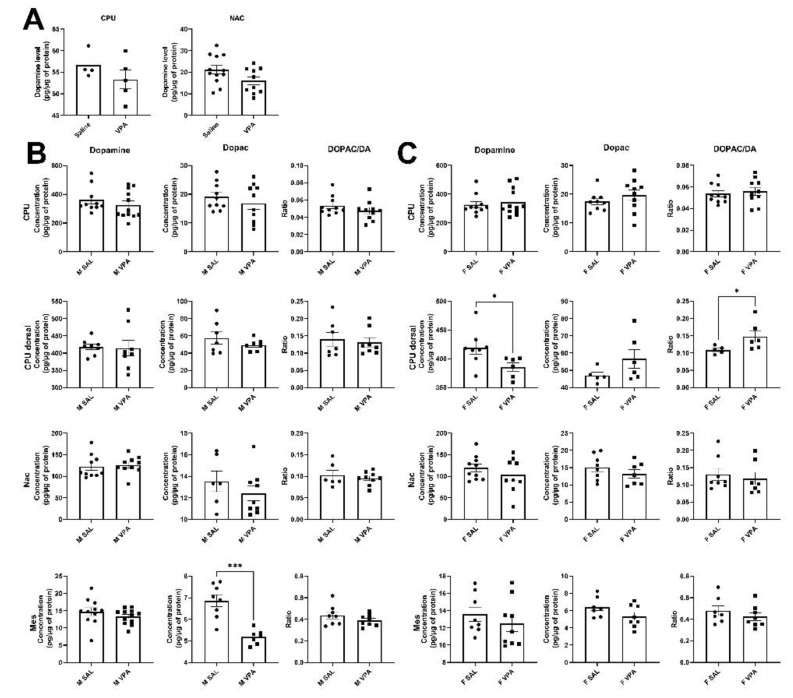
Dopamine and metabolite levels in various brain regions as measured by HPLC (**A**) and mass spectrometry (**B,C**). (**A**) No alterations were found in the levels of dopamine within the striatum or nucleus accumbens as measured by HPLC, (Saline (*n =* 12) and VPA (*n =* 10) in total). (**B**) A decrease in DOPAC was found in the mesencephalon of VPA males. (**C**), A slight decrease in the levels of dopamine was found in the group of VPA females within the dorsal striatum accompanied by an increase in the DOPAC/DA ratio. Abbreviations: ventral mesencephalon (Mes), Nucleus accumbens (Nac), striatum (CPU). Male saline (*n =* 18), male VPA (*n =* 20), female saline (*n =* 18), female VPA (*n =* 18) in total. All data are expressed as means ± SEM; the Student’s *t*-test (HPLC in A) or Mann–Whitney (spectrophotometry in **B**,**C**) test were performed (* *p* < 0.05; *** *p* <0.001).

**Figure 3 biomolecules-12-01691-f003:**
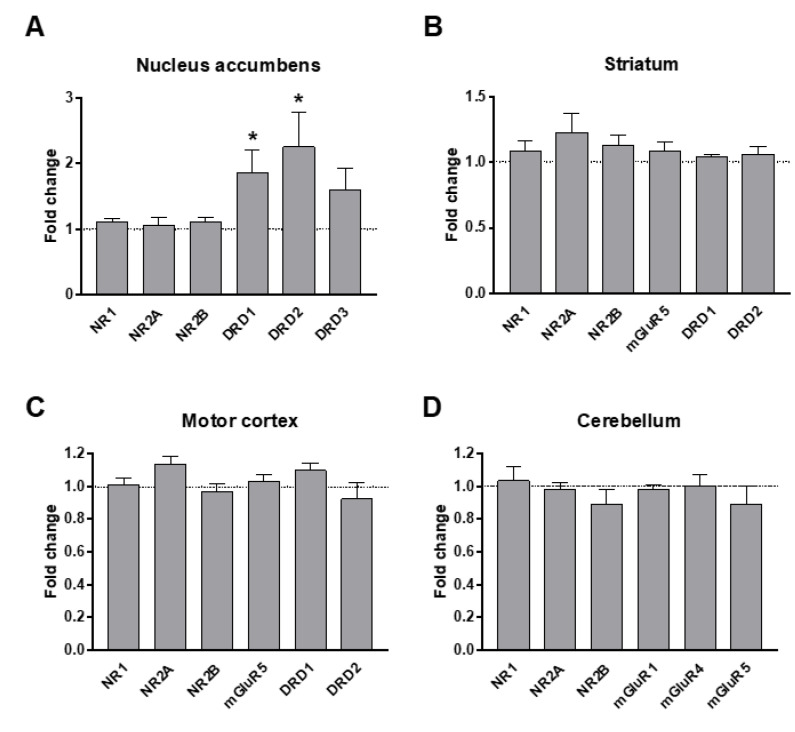
mRNA levels of glutamate and dopamine receptors in response to VPA prenatal exposure. (**A**) Note that D1R and D2R mRNAs are significantly increased in VPA mice compared to saline, in the nucleus accumbens. (**A**–**D**) No other alteration is observed in mRNA levels of NR1, NR2A, NR2B, mGluR1, mGluR4, and mGluR5. All data are expressed as mean ± SEM; Student’s *t*-tests were performed (* *p* < 0.05). Saline males (*n =* 9); VPA males (*n =* 8).

**Figure 4 biomolecules-12-01691-f004:**
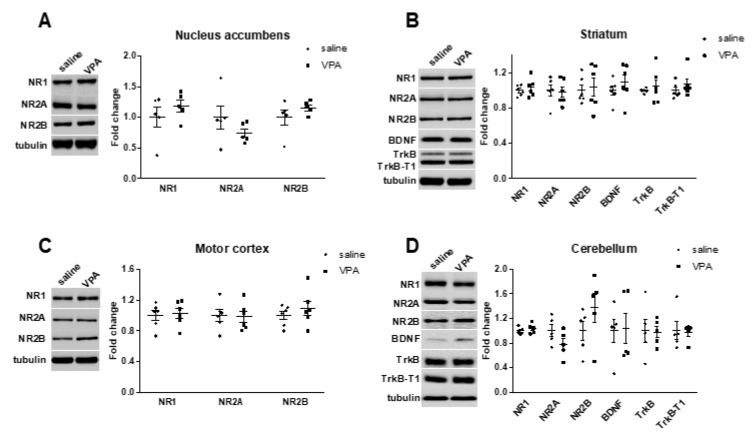
Glutamatergic-related proteins are not altered in several brain regions in response to VPA prenatal exposure. (**A**–**D**) No change was found in the expression levels of the NMDAR subunits NR1, NR2A, and NR2B within several brain regions. (**B**,**D**) No change was found in the expression levels of BDNF, its receptor TrkB or the truncated isoform TrkB-T1 in the striatum and the cerebellum. All data are expressed as means ± SEM; Student’s *t*-tests were performed. Saline males (*n =* 5 to 6), VPA males (*n =* 5 to 6). Figures are representative immunoblots.

**Figure 5 biomolecules-12-01691-f005:**
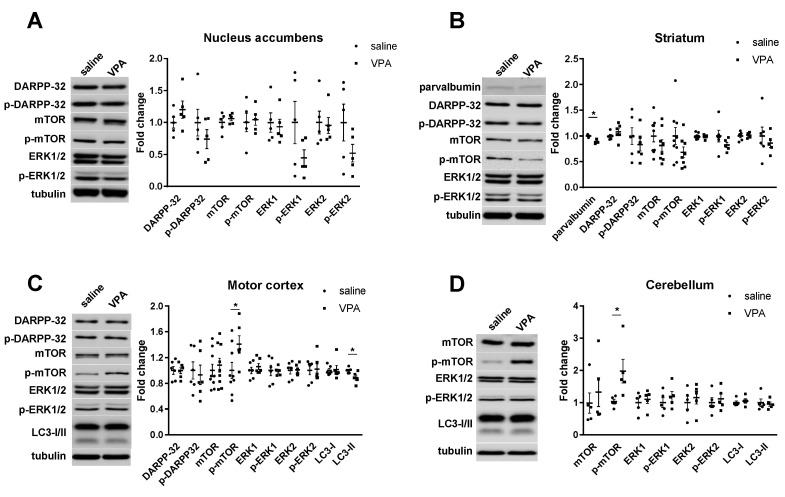
Protein levels of signaling proteins after VPA prenatal exposure. (**A**)**,** No change was found in the expression levels of several phosphorylated and non-phosphorylated proteins in the nucleus accumbens. (**B**) Significant decrease in parvalbumin expression levels was found in the striatum. (**C**,**D**) Significant increase in phosphorylated mTOR protein levels were found within the motor cortex and the cerebellum. All data are expressed as means ± SEM; Student’s *t*-tests were performed (* *p* < 0.05). Saline males (*n =* 5 to 9), VPA males (*n =* 5 to 9). Figures are representative immunoblots.

## Data Availability

Detailed statistical analysis and raw data can be obtained upon request to the authors.

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
