# Peer review of "The State of the Dopaminergic and Glutamatergic Systems in the Valproic Acid Mouse Model of Autism Spectrum Disorder"

_biomolecules, 2022, doi:10.3390/biom12111691_

Round 1

Reviewer 1 Report

General comments:

In the submitted MS the authors report the neurochemistry of the dopaminergic system in the striatum and nucleus accumbens, including expression levels of relevant dopamine receptors, they report glutamate receptors on the level of expression and of proteins and include the respective signal transduction pathways on protein levels in a behaviorally well characterized neurodevelopmental model of autism spectrum disorder (ASD). The extensive characterization is the strength of the submitted MS and deserves reporting to the community, even if there is no spectacular imbalance to report in the glutamate and dopamine neurotransmission which considered a main component of ASD pathophysiology.

However, the discrepancy between on the one hand the summary of their findings claiming in general no alterations in the descendants after valproic acid administration to the dam except an upregulation of dopamine receptors in the nucleus accumbens and on the other hand the presentation of upregulation of indicators of neuronal signaling in motor cortex and cerebellum in the figures of the MS without any mentioning in the Results or in the Discussion is rather surprising and should be corrected (see specific comments below).

Specific comments:

Page 12, reference 22:
This paper is not cited in the text and the reference should be removed

Page 2, line 68/69:
Blundell, J. et al. (2010) does not report data on glutamatergic receptor agonists improving ASD symptoms

Page 2, line 82/83:
Sears,L.L. et al. (1999) do not mention dopamine or dopaminergic connectivity with a single word.

Page 3, lines 120/121, 127 and 140:
What is the distribution of the sex of mice in the stereology data and in the quantitative immunoblots? The context between the number of animals given in the Materials and Methods section and the single points shown in the panels of Fig. 4 and 5 is unclear, and no information is given on the number of animals in the legends of the figures 3-5!

Page 4, lines 155-161:
There is information missing on antibodies to the following proteins shown in immunoblots:
TH, LC3-I/II, BDNF, TrkB, TrkB-T1.
On the other hand, the antibody information on CAMKII, GluR2, Shank3, mGluR5 and GAPDH seems redundant, since none of these proteins is shown in figures or mentioned in the text.

Page 4, line 188:
Please explain what “CAN” means.

Page 5, line 236/237 (legend to Fig.1):
Please explain, what particular immunostaining or immunoblot (representative?) is shown to the left of the graphs with the single points.

Page 4, line 181:
Please describe, how tissue was homogenized and extracted before injection into the HPLC-systems, since reference Solinas, M. et al. (2009) did not report analysis of tissue samples.

Page 5, para 3.2 Dopamine and metabolites levels…, Fig.2:
Based on the information in “Materials and Methods” levels were given in amount per wet weight. Please use the same units in the experiments with HPLC/electrochemical detection (what does ng/uL mean?) and HPLC/mass spectrometry; about 1000 pg/µg = 1000µg/g is actually considerably lower than the dopamine tissue level generally published in the literature! Please comment on that.

Page 7, line 272/273:
Referring to “…decreased dopamine levels as reported here”: There are no reduced dopamine levels reported in the accumbens (see Fig.2B).

Page 7, line 280:
Please explain what you mean by “M1/M2 motor cortices” in the text which refers to “Motor cortex” in Fig.3C.

Page 7, line 282:
Protein levels of NR1, NR2A, NR2B are not shown in Fig. 4, but Fig.5.

Page 8, Figures 4 and 5:
Legends seem to be mixed up; furthermore, following the flow of the text, the figure with the protein levels of NR1, NR2A, NR2B should be presented first, then the figure with protein levels of signalling proteins should be presented. Please adjust the text in line 293-286 and line 302-305 accordingly.
Please refer to the asterisk in the parvalbumin quantification in the striatum.

Page 9, line 306-309:
Why do you write “
We found no modifications in the levels of any of these proteins whether in their phosphorylated or non-phosphorylated forms within the striatum, the nucleus accumbens, the M1/M2 motor cortices or the cerebellum” if there is a significant increase in the levels of p-mTOR in the motor cortex and cerebellum with actually no overlap of the single values in the latter? Please modify this part of the Results, refer to the asterisk in the p-mTOR quantification in the respective legend and insert a passage pertinent to that in the Discussion.

Page 9, line 308:
Please explain what you mean by “M1/M2 motor cortices” in the text which refers to “Motor cortex” in Fig.4C.

Page 9, line 311:
Please correct the wording in “In addition, PV as shown to be decreased in ASD patients and in ASD animal models. ”

Page 9, line 313:
Why do you write “(data not shown)” is there is a parvalbumin band and quantification in the panel for the striatum?

Author Response

Answer to the reviewers.

We would like to start by sincerely thanking the reviewers for their thorough reviews of the submitted manuscript.  We are pleased to read that reviewer 1 found that our paper reports “extensive characterization (which) is the strength of the submitted MS and deserves reporting to the community” while reviewer 2 congratulated us for “the great work on getting these important data”.

However, both reviewers also raised several concerns and we strived to answer them and correct the manuscript accordingly as detailed below.

Reviewer 1

Discrepancy between the summary and the results/discussion section.

*** We agree with this reviewer that consistency in the description of the results is required. We have now made several changes to theses sections accordingly and put more emphasis on the facts that VPA prenatal exposure indeed alters several biological parameters within the dopamine transmission system.

Page 12, reference 22: This paper is not cited in the text and the reference should be removed

***Reference 22 was initially cited on page 11 line 407.

Page 2, line 68/69: Blundell, J. et al. (2010) does not report data on glutamatergic receptor agonists improving ASD symptoms

***We apologize for this mistake in our reference list. This has now been corrected in the revised version of the manuscript lines 67-70. The sentence was rewritten, and three appropriate references were added.

Page 2, line 82/83: Sears,L.L. et al. (1999) do not mention dopamine or dopaminergic connectivity with a single word.

***We apologize for this mistake in our reference list. This has now been corrected in the revised version of the manuscript lines 81-84 with the new references 31-33.

Page 3, lines 120/121, 127 and 140: …Distribution of the sex of mice… Number of animals given in the Materials and Methods section and the single points shown in figures.

**** We do understand difficulties in evaluating animal number in total. This is mainly due to the facts that sometimes a single animal is needed to harvest several brain regions (striatum, accumbens, mesencephalon, cerebellum, motor cortex) while an additional animal is sometimes needed when two regions are overlapping (striatum versus dorsal striatum) and are studied within the same procedure as was the case with the spectrometry analysis for example. We have now mentioned the number and sex of animals per procedure (and not in total as this can be misleading) and in each figure legend.

Page 4, lines 155-161: There is information missing on antibodies...

**** We apologize for these discrepancies and thank the reviewer for his attention. This has now been corrected in the revised manuscript (lines 207-212) (please note also that the materials and methods section has been reorganised.

Page 4, line 188: Please explain what “CAN” means.

*** This was a bad autocorrect. It should have read ACN for acetonitrile. This has now been corrected in the revised version of the manuscript, line 165.

Page 5, line 236/237 (legend to Fig.1): Please explain immunoblots shown to the left of the graphs.

*** This has now been better detailed in each figure legend of the revised version of the manuscript. All original immunoblots are provided in supplementary materials.

Page 4, line 181: Please describe, how tissue was homogenized and extracted before injection into the HPLC-systems, since reference Solinas, M. et al. (2009) did not report analysis of tissue samples.

****A description of the extraction protocol is now added in the revised version of the manuscript, lines 156-157.

Page 5, para 3.2 Dopamine and metabolites levels…, Fig.2: Based on the information in “Materials and Methods” levels were given in amount per wet weight. Please use the same units in the experiments with HPLC/electrochemical detection (what does ng/uL mean?) and HPLC/mass spectrometry; about 1000 pg/µg = 1000µg/g is actually considerably lower than the dopamine tissue level generally published in the literature! Please comment on that.

***Most data using spectrophotometry analysis express the results in ng-pg/uL although we do agree with the reviewer that by doing so it is difficult to compare results between different studies as dilutions and procedures vary. We are now expressing the HPLC and the spectrophotometry data as pg of DA per ug of protein.

-The values of DA levels measured by HPLC are within range of previously reported values (see for instance Mounsey RB et al., Exp Neurol, DOI: 10.1016/j.expneurol.2015.07.024 and Noelker C et al., Scientific Reports, DOI:10.1038/srep01393).

-The values of spectrophotometry data are also in the range of previously reported values (see for instance Wojnicz A et al., Clinica Chimica Acta, DOI: 10.1016/j.cca.2015.12.023).

Statistical analyses were performed again on these new values.

HPLC measurements were performed on males only, as also performed in all other experiments. Spectrophotometry measurements were performed on males and females and results were expressed separately in figure 2. This is also now more clearly stated in the methods section.

Page 7, line 272/273: Referring to “…decreased dopamine levels as reported here”: There are no reduced dopamine levels reported in the accumbens (see Fig.2B).

*** This has now been corrected in the revised version of the manuscript.

Page 7, line 280 and Page 9, line 308: Please explain what you mean by “M1/M2 motor cortices” in the text which refers to “Motor cortex” in Figures.

*** All reference to “M1/M2 motor cortices” was replaced by “motor cortex” as also expressed in figures.

Page 7, line 282 and Page 8, Figures 4 and 5: Protein levels of NR1, NR2A, NR2B are not shown in Fig. 4, but Fig.5.

*** Legends of figures 4 and 5 have now been corrected in the revised version of the manuscript. Additionally the order of figures 4 and 5 has been inverted as rightfully requested by this reviewer.

Page 9, line 306-309: Why do you write “We found no modifications in the levels of any of these proteins whether in their phosphorylated or non-phosphorylated forms within the striatum, the nucleus accumbens, the M1/M2 motor cortices or the cerebellum” if there is a significant increase in the levels of p-mTOR in the motor cortex and cerebellum with actually no overlap of the single values in the latter?

*** We agree with the reviewer that a better consistency in the description of the results was needed. This has now been corrected within the various sections of the revised manuscript.

Page 9, line 311: Please correct the wording in “In addition, PV as shown to be decreased in ASD patients and in ASD animal models.”

*** There was a “w” missing before the “as”. This has now been corrected in the revised version of the manuscript.

Page 9, line 313:
Why do you write “(data not shown)” is there is a parvalbumin band and quantification in the panel for the striatum?

***Parvalbumin data is indeed included within figure 5 and this has now been corrected in the revised version of the manuscript.

Reviewer 2 Report

Great work on getting these important data.  Please use the following comments/suggestions if you revise your paper

1. Abstract – delete: “that we had previously characterised 16 at the behavioral and cellular levels.”

2. Introduction – the valproate model needs to be described and referenced – how is it created, what is its phenotype, and how does it stack up as a model validity wise

3. Statistics – the symbols for stat. significance need to be defined in the figure captions.

4. Dopamine and Glutamate:

     A.  Please discuss the possibility of compensatory effects in this model – e.g., that dopamine and glutamate systems are not much altered because other biochemisty is impacted that protects it from insult – and/or related discussion please.

     B. Please spend a little time in the Discussion on summarizing what biochemical changes, if not dopamine and glutamate, have been seen and how reliable are those findings

5. English – please ask someone to help with this throughout the ms.

Author Response

Answer to the reviewers.

We would like to start by sincerely thanking the reviewers for their thorough reviews of the submitted manuscript.  We are pleased to read that reviewer 1 found that our paper reports “extensive characterization (which) is the strength of the submitted MS and deserves reporting to the community” while reviewer 2 congratulated us for “the great work on getting these important data”.

However, both reviewers also raised several concerns and we strived to answer them and correct the manuscript accordingly as detailed below.

Reviewer 2

  1. Abstract – delete: “that we had previously characterised at the behavioral and cellular levels.”

*** This has now been deleted in the revised version as requested.

  1. Introduction – the valproate model needs to be described and referenced – how is it created, what is its phenotype, and how does it stack up as a model validity wise.

*** A better and detailed description  of the VPA model has now been added in the revised version of the manuscript, in the introduction section lines 105-115 and in the materials and methods section lines 125-128.

  1. Statistics – the symbols for stat. significance need to be defined in the figure captions.

*** This has now been added in all figure legends as also requested by reviewer 1.

  1. Dopamine and Glutamate:

***As also requested by reviewer 1 we have now made changes in the results and discussion sections to better describe our findings and how they relate to previous reports. We are also referring to our recent review entitled “Cerebellar and Striatal Implications in Autism Spectrum Disorders: From Clinical

Observations to Animal Models. Int J Mol Sci. 2022 Feb 18;23(4):2294. doi: 10.3390/ijms23042294. In this review we detail dopamine, glutamate and GABA implications in ASD. We don't wish to go further here in these discussions as they may appear too speculative and not directly related to our presented data.

  1. English – please ask someone to help with this throughout the ms.

***The manuscript underwent English editing by a specialized service.

Round 2

Reviewer 2 Report

Thank you for the thoughtful revisions.